



# Chemical ozone loss and chlorine activation in the Antarctic winters
# 2013–2020
Raina Roy[1,2], Pankaj Kumar[1], Jayanarayanan Kuttippurath[1], Franck Lefevre[3]
[1] *CORAL, Indian Institute of Technology Kharagpur, Kharagpur–721302, India.*
[2] *Department of Physical Oceanography, Cochin University of Science and Technology, Kochi, India.*
[3] *LATMOS/IPSL, Sorbonne Université, UVSQ, CNRS, Paris, France*
*Correspondence to*: Raina Roy (rainaroy2105@gmail.com)
**ABSTRACT**
Since its discovery in 1985, the formation of ozone holes in the Antarctic and the resulting ultra-violet (UV)
radiation reaching the planet's surface has been a source of major concern. The annual formation of ozone hole in
the austral springs has regional and global climate implications. Ozone depletion episodes can change precipitation,
temperature, and atmospheric circulation patterns, affecting the surface climate primarily in the southern
hemisphere (SH). Therefore, the study of ozone loss variability is important to assess its consequential effects on
the climate and public health. Our study examines and quantifies the ozone loss and its cycle for the past 8 years
in the Antarctic using satellite measurements (Microwave Limb Sounder on Aura). We observe the highest ozone
loss (3.8–4.0 ppmv) in spring 2020 followed by 2016. The high chlorine activation (2.3 ppbv), stable polar vortex
and extensive areas of polar stratospheric clouds (PSCs) (12.6 Million Km²) favored the large ozone loss in 2000.
The spring of 2019 also witnessed a moderately high ozone loss, although the year was marked by a rare minor
warming in mid-September. Relatively smaller ozone loss (2.4–2.5 ppmv) was present in 2017 and 2015. It was
mainly due to reduced chlorine activation and relatively higher temperature in these winters. Additionally, the
chlorine activation in 2015 (1.95 ppbv) was the lowest and the wave forcing from the lower latitudes was very high
in 2017 (up to -60 Kms⁻¹). The analysis shows significant interannual variability in the Antarctic ozone as for the
immediate previous decade. The study helps to understand the role of the dynamics and chemistry in the inter-
annual variability of ozone depletion for the years.

**Keywords**: Antarctica; Ozone loss estimates; Polar Vortex; climate Change; Model simulations
**Short title:** Antarctic ozone loss in 2013–2020





## INTRODUCTION

An important event in the Antarctic stratosphere during the austral spring that has caught global attention ever since its discovery in the 1980s is the Antarctic ozone hole (Farman et al., 1985). The chlorine free-radicals released from the chlorofluorocarbons (CFCs) and other ozone-depleting substances (ODSs) activate the catalytic cycles that led to severe ozone loss (e.g. Stolarski and Cicerone, 1974; Rowland et al., 1976). The extreme cold conditions that prevail in the poles facilitate the formation of Polar Stratospheric Clouds (PSCs), which serve as the activation surface for the ODSs. Apart from these, the relatively stable Antarctic polar vortex also contributes significantly to the formation of ozone holes annually (Solomon et al., 2014). Since the discovery of ODSs in the 1970s from anthropogenic activities, the ozone loss continued to rise and reached its worst phase in the late 1980s and early 1990s (WMO, 2014; Kuttippurath et al., 2015). The growth in the ODSs was curtailed after the enactment of the Montreal Protocol in 1987. Ratifying of the environmental treaty led to a stabilisation of the loss from the late 1990s to the early 2000s in the Antarctic. Despite this, there was no significant increase in total column ozone during those times (e.g. Weatherhead et al., 2000; WMO, 2007; Angell et al., 2009). Beyond 2000, significant recovery trends in the ozone were presented with evidence from both ground or satellite observations (e.g. Yang et al., 2008; Salby et al., 2011; Solomon et al., 2016; Chipperfield et al., 2017; Kuttippurath and Nair, 2017; de Laat et al., 2017; Pazmiño et al., 2018; Wespes et al., 2019). A reduction in the saturation of ozone loss over the period 2001–2017 was also observed in the Antarctic and thus confirming the positive ozone trends in the region (Kuttippurath et al., 2018).

The rate of ozone loss in the Antarctic is ascertained by a combination of both polar vortex dynamics and chemistry of the winters. Analyses performed by Nedoluha et al. (2016); Strahan et al. (2014); and Strahan et al. (2018) confirmed the importance of chlorine concentration for the formation of ozone holes. They observed a significant reduction in the amount of chlorine free radicals in the twenty-first century and thus, a consistent increase in the corresponding ozone concentration. On the contrary, the longer lifetimes of ODSs (SPARC, 2016), the non-compliance to the protocol and the recent emissions of the ODSs such as CFC-11, brominated ODSs and several other short-lived ODSs (e.g. Maione et al., 2014; Rigby et al., 2017; Montzka et al., 2018; Dhomse et al., 2019), and the increasing emissions of nitrogen oxides (e.g. Butler et al., 2016; Maliniemi et al., 2021) all of which subsequently decelerated the recovery rate of ozone. An evident manifestation of the influence of dynamic variability in the Antarctic polar vortex is found in the years such as 2000, 2002 and 2004 (Langematz and Kunze, 2006). Thus, the polar vortex variability is an important factor that controls the rate of ozone depletion (e.g.



Stolarski et al., 2006; Zhang et al., 2017). Therefore, a combined action of vortex dynamics and stratospheric chemistry can only lead to significant changes in the Antarctic ozone concentration. The study of annual ozone loss is important as the changes in ozone directly and indirectly affect the climate of southern hemisphere (SH), precipitation patterns, and the circulation in austral summer (e.g. Polvani et al., 2011; Bandoro et al., 2014; Damiani et al., 2020)

Here, we present the long-term analysis of ozone loss for the winters 2013–2020 considering the chemical and dynamical characteristics of the periods prior to and of peak ozone loss. Although a few of the years have been studied individually, the long-term analysis helps in better understanding the evolution of the springs/winters (e.g. WMO, 2015; Krummel et al., 2016; Wargan et al., 2020; Klekociuk et al., 2021). The dynamics of these winters are studied using different meteorological parameters. The study offers a high-resolution analysis of the interannual variability of ozone loss at various altitudes using the data obtained from the AURA microwave Limb Sounder (MLS) (Froidevaux et al., 2008; Santee et al., 2008b). The ozone loss is calculated using the passive tracer simulated by the REPROBUS (Reactive Processes Ruling the Ozone Budget in the Stratosphere) chemical transport model (CTM) (Lefèvre et al., 1994). Therefore, we use a single data set and the same method to estimate ozone loss for all eight years to assess the interannual variability, which would make the comparisons among the winters and assessment of polar winters meaningful and coherent.

**DATA AND METHODS**

We have analysed the meteorology of the winters from 2013 to 2020 using the Modern-Era Retrospective analysis for Research and Applications (MERRA-2) data (Gelaro et al., 2017). The MERRA 2 data are available for 42 pressure levels with a spatial resolution of 0.5° x 0.625°. The nature of austral springs is studied by the examination of four parameters such as the polar cap temperature, zonally averaged from 60° to 90°S at 100 hPa, the minimum polar cap temperature at 10 hPa, the area of polar stratospheric clouds at 460 K, and the mean heat flux averaged from 45° to 75°S. Besides, the MERRA 2 dataset is also employed to analyse the vertical evolution of temperature averaged from 60° to 90°S.

We calculate the ozone loss using the REPROBUS model simulations. The model simulates a passive tracer identical to ozone to estimate the loss in each year. It is a three-dimensional model driven by the European Centre for Medium-Range Weather Forecasts (ECMWF) operational analyses. The analysis is performed over the altitude range 1000 to 0.01 hPa (130 levels) (Dee et al., 2011). In the model, the advection is performed by the winds on





the hybrid sigma pressure coordinates and the trace gases are advected by a semi-lagrangian technique (Williamson
and Rasch, 1989). For our study, the passive tracer identical to the ozone was initialized on July 1 of each year and
continued until the end of November. Then the loss is computed by subtracting the measured ozone from the
modelled passive tracer. The loss in each day is estimated inside the polar vortex as it is more prevalent there and
thus, the polar vortex edge is calculated using the equivalent latitude (Nash et al., 1996; Müller et al., 2005).
The ozone and the chlorine monoxide data are taken from the AURA microwave limb sounder (MLS). The MLS
data of version 4.2 is used. The data have a vertical resolution of 2–3 km, a vertical range of 261–0.02 hPa and an
accuracy of 0.1–0.4 ppmv. Along with this, the ClO measurements are obtained at 640 GHz and are about 3–3.5
km over 147–1 hPa, and the accuracy of measurements is about 0.2–0.4 ppbv. These measurements have a latitude-
dependent bias of around 0.2–0.4 ppbv, depending on vertical height (Livesay et al., 2013).
**RESULTS AND DISCUSSION**
**Meteorology of the winters**

Figure 1 shows the meteorology of the winters as illustrated, with the polar cap temperature (60–90°S) at 100 hPa,
minimum temperature averaged at 50°–90° S at 100 hPa, PSC area (460 K) and mean heat flux from 45° to 75°S
at 100 hPa. The first panel shows the mean temperature (60°–90°S) at 100 hPa, the different coloured lines represent
individual years. The lowest temperature for all the years is observed during August. The temperature begins to
decrease since the beginning of winter (June) and reaches the lowest in August. It is observed from Fig. 1 that all
the years have a temperature in the order of 195–208 K during this period. In the years 2013, 2014, 2015 and 2020
the temperature reduces below 195 K (PSC formation threshold) for a short while in August. However, the
temperature in 2019 shows a sudden rise from late August (202 K) to mid-September (218 K), indicative of the
occurrence of a Sudden Stratospheric Warming (SSW) event. This event has been reported in many of the previous
studies and has been described as a minor Warming (mW) (e.g. Shen et al., 2020a,b; Yamazaki et al., 2020). The
temperature in August 2017 is also seen to be higher than the other years but lower than that in 2019. There is a
visible rise in temperature in the beginning of austral spring. However, temperatures persist below 195 K during
early September in 2015. The lowest temperature range during the spring-winter is present in 2015. The
temperatures in 2020 also follow this range closely, as depicted in Fig 1.





Second panel of Fig 1 shows the minimum temperature during the day averaged in the latitudes surrounding the
polar cap. The minimum temperature for all the winters is found to be lower than the PSC formation threshold.
This continues for the early spring for all the years except 2019 and the minimum value rises soon after and is
higher than 195 K in late spring. The minimum temperature reached their lowest during the day for most days
during the spring and winter of 2015, 2018 and 2020. Thus, ideal conditions for the formation of PSCs are found
for the majority of winter and spring in all 8 years. Therefore, the PSC area grows since the beginning of austral
winter and is observed to be the highest in August (up to 28 million $Km^2$). Corresponding to the periods of longest
minimum temperature, the PSCs are estimated to persist until early November in 2015, 2018 and 2020, but are
short-lived in 2017 and 2019. As the mean temperature peaks in early to mid-September in 2019, the PSC area
drops and diminishes by late September. They dissipated by mid-October in 2017.
A major factor affecting the strength of the polar vortex is the tropospheric forcing. The strength of these forcings
is highly reduced in the Antarctic except for a few winters. In the study of Zuev et al. (2019), the strengthening of
the Antarctic polar vortex in winter and spring is due to the seasonal temperature variations in the subtropical lower
stratosphere. The last panel of Fig 1 shows the tropospheric forcing for the years. The heat flux forcings averaged
between the adjacent mid-latitude and higher latitudes is found to be directed southwards particularly in late winter
and early spring. The years 2019 and 2017 are characterised by very wave forcings, as shown by the flux (from -
40 to -50 $Kms^{-1}$) in the figure. The study conducted by Klekociuk et al. (2020) reported that the easterly phase of
QBO in 2017 also favoured the influence of enhanced wave activity on the polar vortex. The studies of Milinevsky
et al. (2019) and Evtushevsky et al. (2020) also imply similar results about both winters. The zonal average of heat
flux stays between -30 $Kms^{-1}$ and 10 $Kms^{-1}$ for much of the winter and the flux increases as the spring approaches.
The magnitude of these forcings is limited for the years 2015 and 2020.
**Temporal evolution of temperature with altitude**
Figure 2 represents the temporal evolution of zonal mean (60°–90°S) temperature profile in the Antarctic for the
years 2013–2020. The coloured contours show the growth of temperature across the seasons. The white contour
lines represent 188, 195 and 210 K. The zonal winds (westerlies) are overlaid in the figure using the black contour
lines and the easterlies are shown in red. The higher temperature contours descend to the lower stratosphere towards
the end of austral spring. From the analysis, we observe that the descent to the lower stratosphere is maximum in
2019 during mid-September as a consequence of the minor warming. The temperature contours in the range 250–
265 K extend to slightly below 10 hPa. We also notice a slight reduction in the speed of westerlies during the



period. The zonal mean temperature contours reveal that temperatures below 195 K are found in the lower
stratosphere (100–70 hPa) until mid-October in 2015 and 2020. This implies that the temperature conditions during
these years are the most ideal for the highest ozone loss. Similarly, the area covered by 195 K is also moderately
high in 2013, 2014, 2016 and 2018. However, this is lowest in 2019 and is minimal in 2017. The appearance of
easterlies below the 10 hPa altitude is late in 2015 (late November) whereas in 2019 and 2017 the easterlies appear
as early as late October. Thus, the vortex lasted the longest in 2015.
**Temporal evolution of ozone**

Figure 3 shows the temporal evolution of ozone (in ppmv) from MLS data for the period 2013–2020. It is observed
from previous studies that the ozone loss is maximum in the lower stratosphere in all years (Solomon et al., 1990).
The ozone concentration reduces in the lower altitudes every year as time progresses (mainly in spring), as
illustrated in Fig 3. Opposed to this, the concentration of ozone in the upper stratosphere increases as time evolves.
The ozone concentration in the years 2013, 2014, 2015, 2016, 2018 and 2020 in the lower altitudes (400–600 K) is
around 0–3 ppmv. Unlike in the cold winters/springs, the concentration of ozone is slightly higher (1.5–2.5 ppmv)
in the same altitude range in 2019. Similarly, in 2017, the ozone concentration in the lower stratosphere is higher
than the previous cold years owing to the higher temperature present in the stratosphere. The lowest ozone
concentration for the altitude range 400–475 K is observed during the years 2018 and 2020. The 0.5 ppmv contour
extends to a level of 475–500 K in both years.

**Chlorine activation and ozone loss in 2013–2019.**

Figure 4 presents the temporal evolution of ClO (right) and ozone loss (left) at different altitudes during the period
of study. The ozone loss is highest in the altitude range 400–550 K or in the lower stratosphere for all years and is
observed in September and October. The loss is less than 1.4 ppmv in the upper stratosphere in all years. The ozone
loss in 2014 and 2015 are almost similar, at about 2.6–3.0 ppmv in the peak ozone loss altitude during September
and October. The ozone loss in 2013 reaches up to 3.2 ppmv in mid-October and is higher than that in 2014 and
2015 (Vargin et al., 2020). The ozone loss reported in Strahan et al. (2018) for 2015 is similar to the very cold
winters in Antarctica and is slightly higher than that found in our study. The preconditioning for ozone loss in 2013
and 2014 is ensured by the high chlorine activation at the same altitude range (Kuttippurath et al., 2015). Among
the three years prior to the period of the highest ozone loss, the chlorine activation reaches the maximum values in





August and September. The chlorine monoxide concentration is found to reach up to 2.2 ppbv in 2013 and 2014, and 2.0 ppbv in 2015. This high chlorine activation remains for almost a month in the peak ozone loss altitude in 2013 and for lesser duration in 2014 and 2015. The austral springs of 2017 and 2018 also show similar value ranges for ozone loss, and the chlorine activation in the order of 1.8–2.2 ppbv last for a period of 15–20 days before attaining the maximum ozone loss.

We observe the ozone loss in spring 2016 to be about 3–3.2 ppmv in September and 3.4 ppmv in October. It is also noticed that the ozone hole, the PSC occurrences, and chlorine activation (more than a month, up to 2.2 ppbv) lasted longer for the year. An extensive ozone hole is found in 2019 and it lasts from late August to mid-November. The ozone loss has exceeded up to 3.6 ppmv since late September. However, the ozone concentration enhanced after the warming and thus the ozone hole size reduced drastically (Fig 3). The chlorine activation is continuous from August to September (above 2.2 ppbv). The ozone loss values in 2019 and 2020 austral spring are observed to be much higher than the other years. Despite the episode of minor warming in 2019 (3.0–3.6 ppmv), the ozone loss was similar to that in 2016. The nature of the spring was similar to previous warm Antarctic years 1988 and 2002. The vortex was short-lived and was highly variable due to increased tropospheric forcing during the year. The ozone loss in 2019 spring was about 3.6 ppmv, which is higher than 2018 ozone loss during the same period (Wargan et al., 2020; Roy et al., 2022). The chlorine activation remained at the peak value (2.0–2.2 ppbv) for several days in August prior to high ozone loss and the spatial distribution (450–550 K) of these high values were the highest compared to all other years. The 2020 ozone loss was very high (up to 4.6 ppmv) and exceeded the maximum ozone loss in 2016. The chlorine activation rose remarkably in the early austral spring (September) (2.0– 2.2 ppbv) and is observed to be similar to that in 2016 during this period. Despite the nominal values of chlorine activation in the 2020 spring, the record values of ozone loss may have resulted from the increased aerosol loading into the Antarctic atmosphere from the Australian bushfires in 2020 (e.g. Stone et al., 2021).

**Interannual variability of ozone loss for the period 2013–2020.**

The interannual variability of ozone loss and chlorine activation are shown in Figure 5. The ozone loss is computed by taking the mean ozone loss (in ppmv) in the altitude range 450–550 K (the peak ozone loss altitudes) for the day 270 to day 300 day (the peak ozone loss period). Similarly, the chlorine activation (in ppbv) is also estimated as a mean of the ClO values over the altitude range (450–550 K) for the day 210 to day 270. The weighted mean of



PSC area is also shown in the figure in black solid line for the years 2013–2020. It is observed that the lowest ozone
loss is estimated for the years 2015 and 2017 as a result of the minimal chlorine activation prior to the events. The
mean ozone loss is 2.4–2.5 ppmv for both the years and the chlorine activation is about 1.95 ppbv (2015) and 2.15
ppbv (2017). However, the PSC area in 2015 (11.9 million Km²) was higher than most of the other cold winters.
Despite the favourable conditions, the low ozone loss was perhaps associated with the relatively weaker chlorine
activation. In the study of Tully et al. (2020), the ozone hole metrics of 2015 is identified to be one of the most
severe and extreme opposed to our results, however, this offset is caused by the difference in the period of
estimation. They have estimated the values of integrated column deficit and ozone hole area metrics for the period
October-December for the corresponding years.

The PSC area in 2017 (10.2 million Km²) was lower compared to this. The lower value of ozone loss in 2017 spring
is also agreed upon in the analysis conducted by Baarthen (2018). Consistent with the results from the column
ozone loss analysis the highest ozone loss is estimated to be in 2020 (3.8 ppmv) spring followed by 2016 (3 ppmv).
The chlorine activation for both years is also higher than a few other cold winters i.e., 2.2 ppbv in 2016 and 2.3
ppbv in 2020. The high ozone loss in 2020 is favoured by the very large PSC area (12.6 million Km²). The 2018
spring was also unique in comparison to the other years of study as a consequence of the high chlorine activation
(2.3 ppbv) and very high PSC area (12.0 million Km²). The chlorine activation was the highest in 2019 (2.35 ppbv),
however, the relatively lower ozone loss in the year is observed to be a direct consequence of the unfavourable
dynamic condition (mW). The PSC area is estimated to be the lowest in 2019 (9.4 million Km²). The ozone loss in
2013 and 2014 were almost similar (2.7 -2.8 ppmv) and the chlorine activation is in the order of 2.2 and 2.25 ppbv
in 2013 and 2014, respectively.


**CONCLUSION**
A major threat that has plagued humanity since its discovery in the early 1980s is the Antarctic ozone hole and its
consequential effects. The ultraviolet (UV) radiation reaching the surface of earth because of the ozone layer
destruction can impact humans and other life forms severely. The effectiveness of Montreal Protocol to control the
emission of these ODSs thus helped in reverting tremendous damage to the life forms on the planet. In addition to
this, ODSs can also cause substantial global warming (Wigley, 1988). Henceforth, a check on their release also
helped in reducing the potential global warming and also on regional climate change. Therefore, the study of



interannual ozone variability is important to understand the ozone related changes in the surface climate. Here, we assess the ozone loss for an 8-year period (2013–2020) in the Antarctic.

The year 2019 had a warm winter with a mW in mid-September. The winter 2017 also showed similar characteristics i.e, there was a sudden increase in temperature during late August, the minimum temperature (about 205 K) was also higher in August than that in other years, and the PSC area showed a sharp decrease towards the end of September in that year. The heat flux magnitude for the year (2017) was also higher than the other winters (up to -60 Kms$^{-1}$). Conversely, the wave fluxes were the lowest in 2015 winter. The temperature and PSC area follow similar temporal evolution in 2013, 2014, 2015, 2016 and 2018. The winter 2020 exhibits a unique meteorology with a long-lasting occurrence of vortex wide PSCs (12.6 million Km$^2$) and has the highest ozone loss (3.8-4.0 ppmv). The lowest ozone loss (2.4–2.5 ppmv) was estimated in 2015. We observed a minimal ozone loss in 2017 and the loss stayed lesser than 2.8 ppmv for most of October and September. Chlorine activation was also below 1.6–1.8 ppbv in August and September for the year. The study, thus, helps in understanding how the chlorine activation and meteorology of the years influenced the variability of ozone during the period. It is also observed that the dynamics and chemistry of the years played their respective roles in the ozone loss process. The winter 2019 is an example of favourable chemistry helping in increased ozone loss though the dynamical conditions were unfavourable.

**Acknowledgements**

We thank the Indian Institute of Technology Kharagpur, for facilitating the study. We acknowledge free use of the MLS data, which are taken from https://disc.gsfc.nasa.gov/. The meteorological data are acquired through https://ozonewatch.gsfc.nasa.gov. The REPROBUS data are acquired through IPSL, http://cds-espri.ipsl.fr/. We thank Cathy Boonne for her help with the model runs, analyses and data transfer, and IPSL for hosting the data.

**Data availability**

The data used in this study are publicly available. The analysed data/codes can also be provided on request.

**Competing Interests**

Jayanarayanan Kuttippurath is a member of the editorial board of Atmospheric Chemistry and Physics.



**Author Contributions**

JK conceived the idea, and JK and RR wrote the original manuscript. The manuscript was subsequently revised with inputs from PK and FL. The model runs and model results were analyzed by FL. The data analyses and figures made by RR and PK. All authors participated in discussions and made suggestions, which were considered for the final draft.





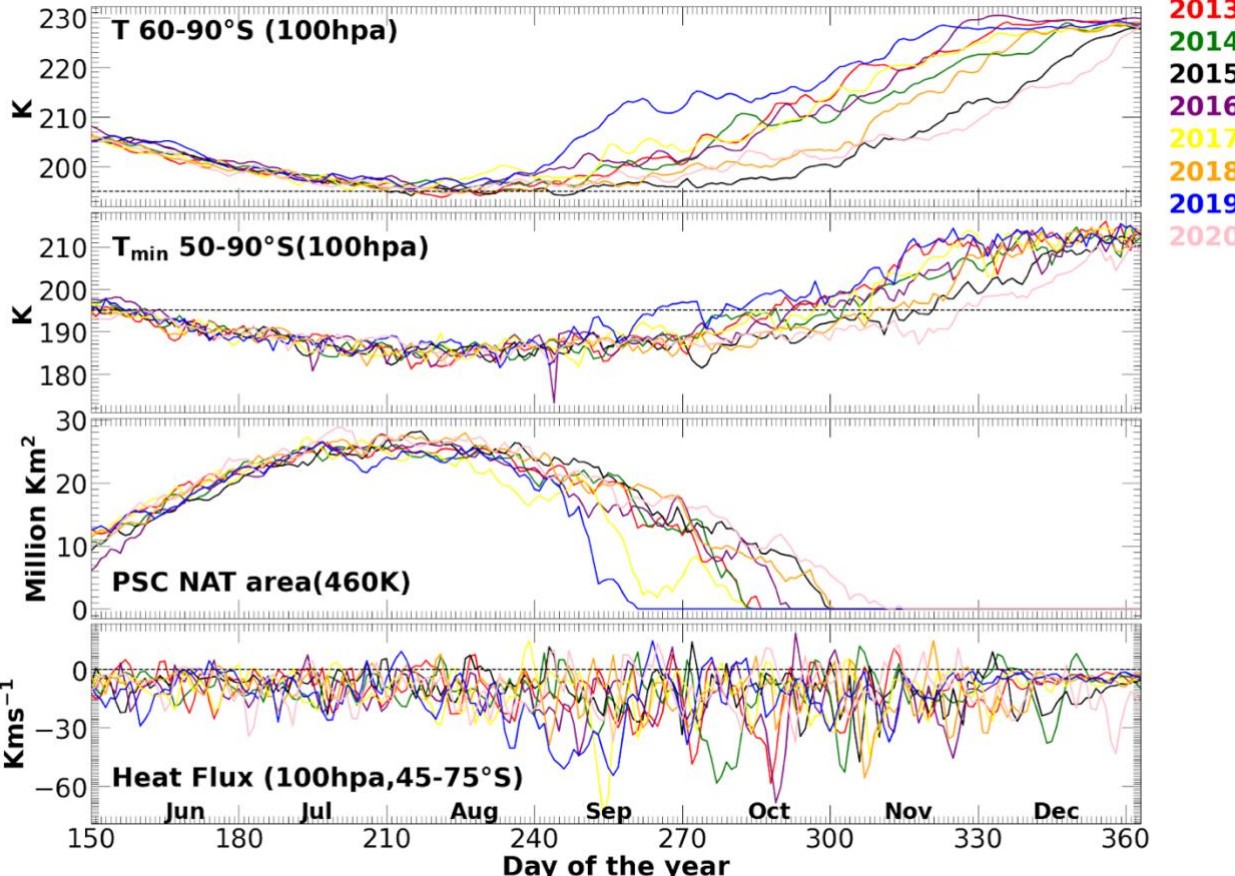

**Figure 1:** Meteorology of the years (2013–2020). First panel shows the zonal average temperature (60°–90°S) at 100 hPa. Second panel (from top) shows the minimum temperature at 100 hPa. The black line in the panels show 195 K (PSC formation threshold). Third panel (from top) shows the PSC area at 460 K and the bottom panel shows the mean heat flux (45°–75° S) at 100 hPa. The black horizontal line in the bottom panel shows zero heat flux.

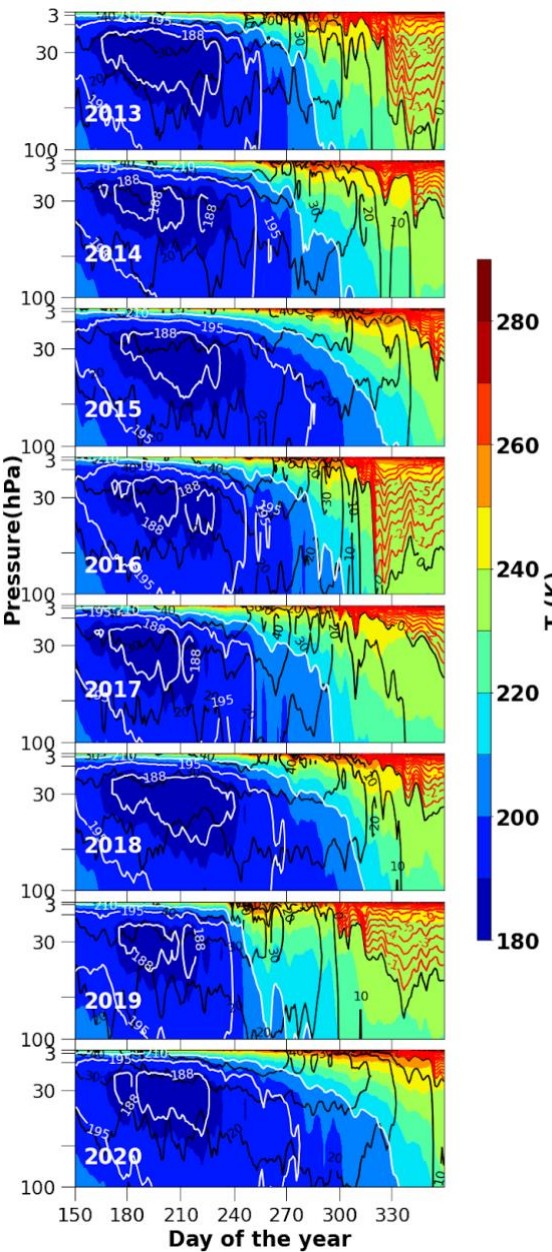

**Figure 2:** Seasonal march of the zonal mean temperature for the period 2003–2020 averaged over the latitudes 60–90°S. The contours show the temperature and the white contours represent specific temperatures such as 188, 195 and 210 K. The zonal wind velocities are overlaid. The black contour lines show the westerlies and the red contour lines show the easterlies.



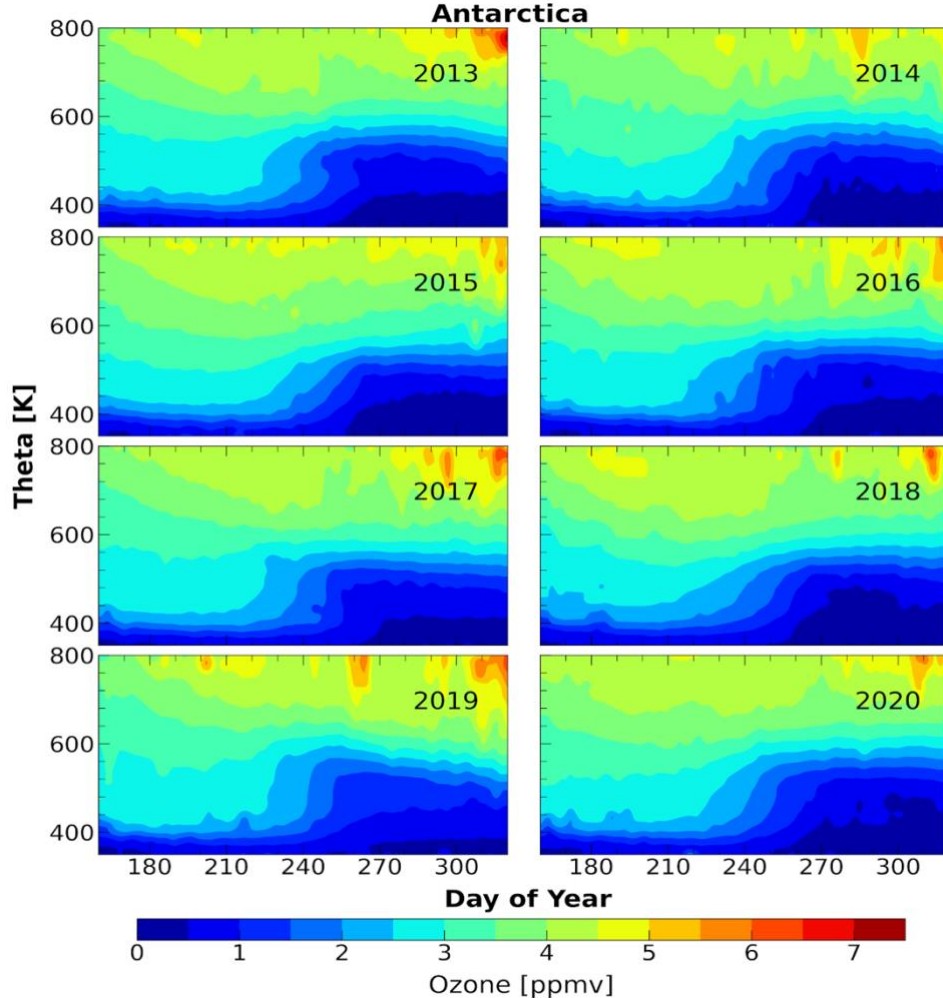

**Figure 3:** Temporal evolution of the vertical profiles of ozone averaged inside the vortex for the winters from 2013 to 2020 in the Antarctic. MLS data are used for the estimation. The temporal evolution is analysed for the period June–November and for the altitude range 350–800 K.



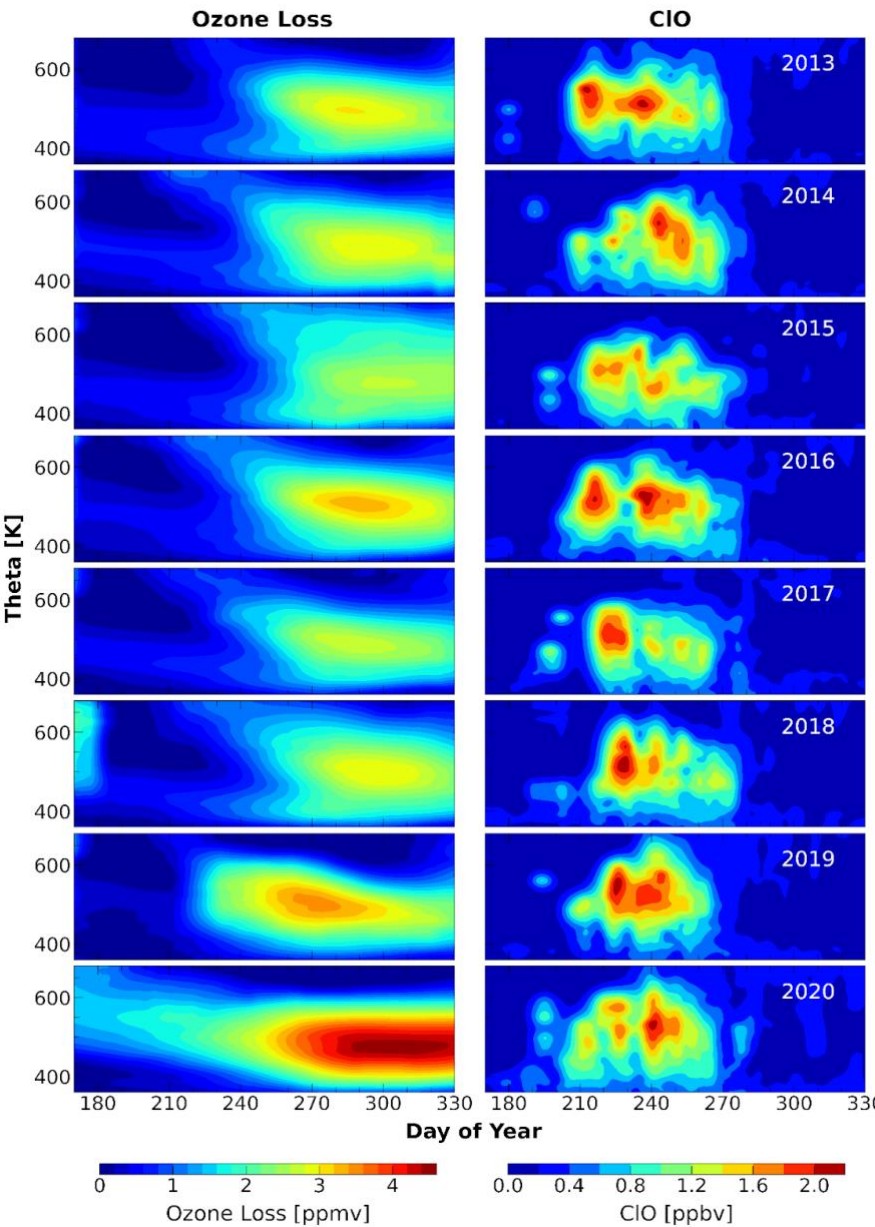

**Figure 4:** Temporal evolution of ozone loss estimated from MLS measurements using REPROBUS passive tracer (left). The MLS ClO measurements for the altitude range 350–700 K for the period 2013–2020. The ozone loss estimates and ClO measurements are selected inside the polar vortex as per the Nash et al. (1996) criterion.



310

311

312

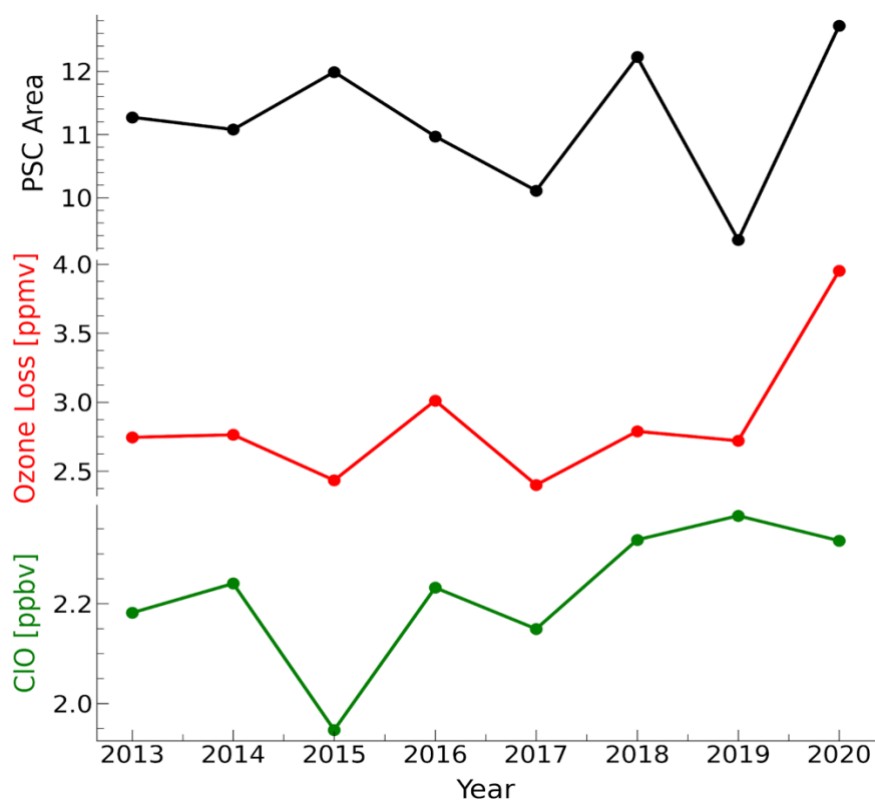

313
314

**Figure 5**: The vortex-averaged ozone loss estimated from the MLS measurements using the passive method, peak ClO measurements, and the weighted average of area of PSC for the period 2013–2020. The mean ozone loss is estimated over the altitude range 450–550 K for the day 270 and day 300 (maximum ozone loss days). The ClO measurements are averaged over the altitude range 450–550 K and for the day 210 and day 270; representing the strong chlorine activation period and altitudes.









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

V08/JK/04112021