# Peer review of "Chemical ozone loss and chlorine activation in the Antarctic winters 2013–2020"

_EGUsphere, 2023_

## Author Comment (AC1)

**REPLIES TO REFEREE #2 COMMENTS**

Thank you very much for your time, review and positive comments on the MS. Please find answers to specific comments below. We do hope that the referee will find the revised version more interesting and recommend a publication very soon.

This paper gives an overview of Antarctic polar vortex ozone loss and meteorological characteristics for a recent eight-year period. Most of these Antarctic winters have been analyzed in previous papers but the period as a whole and the use of model output to diagnose chemical ozone loss is unique. This work doesn't necessarily advance our knowledge of Antarctic ozone hole chemistry or evolution but rather gives an update on the status in recent years. The most interesting result is the model derived ozone loss in Figures 4 and 5, especially in 2020. It would be nice to see how the recent ozone loss compares to earlier years but that may be beyond the scope of the paper. Although the paper doesn't break new ground, I would recommend publication with consideration of the few comments below. There are a number of grammatical errors that should be addressed and I don't have time to list them all. Should include reference to Manney et al., 2020 paper that looked in detail at the 2020 winter compared to several previous winters in the time period shown in this paper.
Done. Please note that Manney et al. (2020) discuss the ozone loss in the Arctic winters. There is no mention for the Antarctic winters discussed here. However, we have cited the work in the Introduction in lines 54. Thank you for understudying.

Also, Ansmann et al., 2022 discuss the 2020 and 2021 Antarctic ozone holes and how they were affected by forest fire smoke.
Done. We have included these studies in discussion in lines 185–186.

Why not use the same reanalysis product for the meteorology analysis as was used to drive the REPROBUS model? Or just use the model meteorology. I'm sure there are differences in the meteorology of MERRA-2 and ECMWF during these years.
Done. In fact, we have taken the PSC area calculated from MERRA as available from https://ozonewatch.gsfc.nasa.gov/. We apologize for the confusion. This is mentioned in lines 64–68.

Line 21: should be 2020, not 2000
Done. This is corrected in line 19.

Line 70: Need to include the Klekociuk et al., 2021 paper in your list of references
Done. We cited and discussed this in lines 54, 121–122.

Line 167: I would remove 'in 2013-2019' from this header. Or at least change 2019 to 2020 since that is the correct end year of the analysis.
Done. Removed, as suggested.
* * *
V06

---

## Author Comment (AC2)

**RESPONSE TO REFEREE #1 COMMENTS**

In the current manuscript, Roy et al. have used the MERR2 reanalysis data set to look at the inter annual variability in the Antarctic stratosphere for 2013-2020. They have also used the MLS O3 measurement combined with a chemical transport model REPROBUS to derive the ozone loss inside the Antarctic region for the past 8 years. The authors have investigated the causes of Antarctic polar ozone loss mainly focusing on different stratosphere and Chlorine activation by looking at the observed ClO evolution from MLS. The current study and methods used here are not NEW and most of the results from the current manuscript are understandable, i.e., most conclusions are consistent with previous studies. There is no exciting result or some interesting sciences to be addressed from this work. However, the paper is well written and organized. The message in the current version is clear. The quantification of chemical ozone loss for the Antarctic polar region under different meteorological conditions is still useful for the atmospheric community. The paper is still publishable at ACP after major revision. Detailed comments are seen below:

Thank you very much for your time, review and positive comments on the MS. Please find answers to specific comments below. We do hope that the referee will find the revised version more interesting and recommend a publication very soon.

Content: In the abstract/introduction, the authors have mentioned that "ozone depletion episodes can change precipitation". How significant causes the precipitation changes due to ozone depletion? Is there any strong evidence to support it?

Done. Yes, there have been studies that showed that the ozone depletion has caused the shift in the westerly jet in the midlatitudes which would lead to increased rainfall there. For instance, please see Kang et al. (2011), Kang et al. (2013), Thompson and Solomon (2002) and Gillett and Thompson (2003). Since it is mentioned in Abstract, we cannot cite references there. Thank you for understanding.

Line 18 in the first page, it is not correct to say "quantifies the ozone loss.. using satellite measurements" because the authors also use the passive ozone from the REPORUBUS model.

Done. This is rephrased as, using satellite measurements and passive ozone simulations, in lines 73-74,79-81.

Following on 2), there are some inconsistencies in the current version. Line 88 says ozone loss is from MLS satellites, Line has caused some confusion about the "ozone loss". Then Line 87 in Page 3 says "we calculate the ozone loss using the REPROBUS model simulations' '. Then the authors mentioned they have calculated the ozone loss in Line 94 Page 4 "the loss is computed by subtracting the measured ozone from the modeled passive tracer....".

Done. The ozone loss is calculated as follows; The observed ozone from MLS is subtracted from the Passive ozone (i.e. the ozone simulated by switching the chemistry off) simulated by the model. This is called passive method, which is a widely used ozone loss estimation method (e.g. Gautail et al., 2005; Kuttippurath et al., 2013). This is mentioned in lines 73-74,79-81.

Line 92 in Page 4 mentioned that the "passive tracer identical to the ozone was initialized on July 1 of each year and continued until the end of November ", but there is a weird large ozone loss in early July in Figure 4. It looks to me that the REPROBUS has not simulated ozone well compared with MLS data most of the years from 2013-2020. The model simulations are even worse for 2018 and 2022 when looking at the first July ozone loss figures. Can you explain why? Is tracer transport/chemistry or other processes causing these discrepancies for 1 July 2018 and 2020.?

Done. Yes, this problem of initialisation and thus the tracer simulations. Therefore, we have removed tracer simulations up to 10 June 2018 and 20 July 2019 and we have not calculated ozone loss. This is mentioned in lines 160–163, and see the revised Figure 4.

Most of the main papers, the authors claimed that "observe" ozone loss. This is only true if REPROBUS can reproduce the observed MLS ozone during the period, but this is not the case (please see 3)).

Done. In fact, we used the model simulations only for ozone loss estimation, not for model-observation comparisons as it is beyond the scope of this paper. The term "observed ozone loss" is used to differentiate it from the modelled ozone loss, as in the previous studies. This is mentioned in line 73-74, 79-81.

The calculation of PSC areas. There is nowhere to mention what the values of HNO3, H2O etc (and where they are from) are used for the PSC diagnostics based on thermodynamic equilibrium.

Done. The PSC area is calculated assuming a fixed profile of nitric acid with a concentration of 4.97 ppt at 460 K. The concentration of water vapour used was about 5 ppm. Further details are provided on https://ozonewatch.gsfc.nasa.gov/meteorology/temp_2022_MERRA2_SH.html) . These are mentioned in lines 69–70.

Line 46 in Page 2, it would be better to specific the region either altitude/regions more specifically for "significant recovery trends in the ozone"

Done. In the lower stratosphere, mentioned in lines 46–47.

Line 49 in page 2, what is the value of "the positive ozone trends"? Based on the current version, have authors also compared the ozone loss over the period of 2013-2020 the period 2001-2017? This would be interesting to know if they have also seen something "A reduction in the saturation of ozone loss" over the period 2013-2017, may be similar or even significant smaller ozone loss rate over 2013-2020 compared to 2001-2017, this will make it robust to say "confirming the positive ozone trends"

Done. We have not presented the trends in ozone here, as it is completely a different topic. Instead, we are looking at the interannual variability of ozone loss and chlorine activation. Also, please note that 8 years of data is not enough for trend analysis and also to make robust statistics. On the other hand, we have compared the ozone loss for all winters using the same criterion in **Figure 5**. Please find the discussion in lines 192–222. A new section (lines 212–222 and **new Table 1** is also added now.

Line 60 in Page 60, why only choose these three years? Better to add other examples here.

Done. Please find the corrected statement in line **141**

Lines 89-90. It looks that REPROBUS is forced by ECMWF operational analyses, which has not nudged the satellite/in-situ observations. Please note that ECMWF operational analysis has changed resolution to 137 vertical levels from 2013, not sure why the paper cites Dee etal. (2011) which is mainly for the description of ERA-Interim reanalysis. Of course, this will have some changes in the model simulations if the authors using the simulation forced by ERA5 (as an example).

Done. Sorry for the mistake. We have rephrased this. Please find it in lines 70-73.

Methods. Since the loss calculation is based on the equivalent latitude (Line 95 in page 4), the authors still use the geographic averaged latitude to do other calculations (for example, temperature, PSC etc..). I would suggest the authors use the same criteria to re-make the figures.

Done. Please note that these types of analyses are mostly performed for polar cap temperature, winds, etc, which is why we have presented the analysis this way. These are also needed to check major and minor warming criteria, as we presented in lines 100–103. In addition, this is also needed to compare with previous studies (e.g. lines 102–103). Therefore, we have kept the original analyses for Figures 1 and 2. However, we respect the referee's comment and we have done the temperature and wind analysis inside the vortex and it is presented in Figure S1 and related text in lines 136–142.

some results should be carefully made, there seems some results mentioned by the authors are not consistent with what has been shown in Figures. For example Line 108 in Page 4, but I can still see the lowest temperature for 2015 occurs in the early September, not in August in the top panel of Figure 1.

Done. This is corrected in lines 104–105.

Why use "growth of temperature", then "descend", "descent". They are improper used for the temperature.

Done. This is made consistent and no such words are used now in the MS.

Line 153, "the vortex lasted the longest in 2015", but looking at Figure1, it seems to me "2020" has the long-lasting cold polar vortex.

Done. Yes, this is rephrased and corrected in lines 96–98, 104–105.

Sometimes there is no explanation for the results shown. For example, Line 171 in Page 6, what causes the still large ozone loss in the upper stratosphere? The authors claimed "The loss is less than 1.4 ppmv in the upper stratosphere in all years".

Done. This is explained in lines 157–158.

For the ozone loss, the authors only look at the ozone loss in different ways (sometimes, they gave largest ozone loss values using different altitudes or averaged different regions or periods). I would suggest the authors add the partical column ozone loss, then make one table to list the partial column ozone loss, peak ozone loss, averaged ozone loss etc, which should make the readers understand the key ozone loss results from 2013-2020.

Done. Please note that Figure 5 is made for the comparison for different winters as it shows the amount of PSC, ClO and ozone loss computed using the same criterion for all winters. However, as suggested, we have also made the partial column loss comparison. Please see the **new Table 1** and related text lines 212–222.

Again, there is inconsistency in the text and the caption of Figures. For example, Line 206 (mean of the ClO values...." and Figure 5 "peak ClO measurements". For Figure 5, there is no explanation why 2015 has the largest PSC areas than other years, which is very hard to see from all the figures including Figures 1 and 2.

Done. These are corrected and explained now in lines 192–193, 197–198. Thank you.

V06

---

## Editor Decision (ED1)

**Comments on Roy et al.: *Chemical ozone loss and chlorine activation in the Antarctic winters 2013–2020**

**General comments**

The track changes document looks incomplete. There seem to be several new pieces of text which are not highlighted in it. In addition, omitted pieces of text are not included in the track changes. This makes the review of the revision complicate and makes it very difficult to judge if the concerns risen by the referees were considered. Please provide complete track change information on the next revision.

In the following, line numbers refer to manuscript version 3.

**Specific comments**

– Referee#1 has asked about your mentioning of the influence of ozone depletion on precipitation. This is mentioned in the abstract, but it does not seem to be discussed in the introduction. Please add with references.

– I agree with the referee that your usage of the term "observed ozone loss" is confusing as you derive it from an observed and a modelled quantity. So there is no direct observation of ozone loss here. Please use a more appropriate term such as e. g. "derived" or "inferred".

– L 69: Please add the URL that you used in your response to the referee to the manuscript or a reference to a peer-reviewed publication to justify the values mentioned.

– L 202: check reference: the author's name in the text does not agree with the list of references. The citation is incomplete.

– Table 1: there does not seem to be a reference to the new Table 1 in the text. Please discuss the information presented in the table and reference it.

– Figure 4: In response to a comment from Referee#1 you omitted the tracer simulations up to 10 June 2018 and 20 July 2019 in the respective figure panels.

What was the reason for the choice of date? How about the initialisation phase for the other years? How about the year 2020 when ozone loss numbers presented in Fig 4 also looked very different at the beginning of the season? The new Figure 4 has a modified colorscale but this feature is still well visible but not commented on. What is the purpose of the isolines highlighted in black in the new version?

– Data availability statement: please add doi information or URL for all publicly available data

– I don't see where/how you addressed referee#2's question about mixing up MERRA-2 and ECMWF analysis data. In Lines 64/65 it is stated that MERRA-2 meteorology is used while lines 72/73 state that the REPROBUS model is driven by ECMWF operational analysis. Referee#2 has asked why you did use the meteorology that drives the model also for your analysis and I do not see that question answered, neither in your response nor in the revised manuscript.

– I don't see a sufficient discussion of the mixing up latitude with equivalent latitude raised by referee#1

– choice of words regarding "temperature growth" etc.: please provide an accurate track of changes made for assessment.

---

## Author Response (AR2)

**RESPONSE TO THE EDITOR AND REFEREE COMMENTS**

General comments: The track changes document looks incomplete. There seem to be several new pieces of text which are not highlighted in it. In addition, omitted pieces of text are not included in the track changes. This makes the review of the revision complicated and makes it very difficult to judge if the concerns risen by the referees were considered. Please provide complete track change information on the next revision:

Thank you very much for your time, review and comments on MS. Please find answers to specific comments below. **Please note that we have given the line numbers of the answers or changes in the MS for each query to identify the respective responses**. Please find the response to queries and the corresponding change in the MS in **blue typeface**. We do hope that the Editor and referees will find the revised version more interesting and recommend a publication very soon.

We have again made the track-change version of MS and all changes shown are in blue typeface. Please note that there were some typos, grammatical corrections and rephrasing sentences, which are not shown in track-change mode.

Content: Referee #1 has asked about your mentioning of the influence of ozone depletion on precipitation. This is mentioned in the abstract, but it does not seem to be discussed in the introduction. Please add with references.

Done. Yes, there are studies discussed about the precipitation and ozone hole connection. This is discussed now in the revised MS. For instance, please see the references Kang et al. (2011), Kang et al. (2013), Thompson and Solomon (2002) and Gillett and Thompson (2003). We have mentioned this and cited the articles in the introduction. Please find l**ines 49-55**.

I agree with the referee that your usage of the term "observed ozone loss" is confusing as you derive it from an observed and a modelled quantity. So there is no direct observation of ozone loss here. Please use a more appropriate term such as e. g. "derived" or "inferred". Line 18 in the first page, it is not correct to say "quantifies the ozone loss.. using satellite measurements" because the authors also use the passive ozone from the REPORUBUS model.

Done. This is rephrased as suggested, "inferred ozone loss". Please find it in l**ine 84**.

L 69: Please add the URL that you used in your response to the referee to the manuscript or a reference to a peer-reviewed publication to justify the values mentioned

Done. We have added the link with the description in the Data and Methods section in **line 75.**

L 202: check reference: the author's name in the text does not agree with the list of references. The citation is incomplete.

Done. We have made the correction in l**ine 205**.

Table 1: there does not seem to be a reference to the new Table 1 in the text. Please discuss the information presented in the table and reference it.

Done. Please find the reference to Table 1 in **line 216.**

Figure 4: In response to a comment from Referee#1 you omitted the tracer simulations up to 10 June 2018 and 20 July 2019 in the respective figure panels. 1 What was the reason for the choice of date? How about the initialisation phase for the other years? How about the year 2020 when ozone loss numbers presented in Fig 4 also looked very different at the beginning of the season? The new Figure 4 has a modified color scale but this feature is still well visible but not commented on. What is the purpose of the isolines highlighted in black in the new version?

Done. We have noticed that the model tracer simulation was not initialised on 1 July (as it should be for all winters) and therefore, there is a bias in the tracer simulations. However, the simulations looked reasonable after these dates in all years. The bias and initialisation problems were identified after comparison with the measurements (measured and simulated ozone, and simulated tracer). Therefore, we did not use the tracer simulations for ozone loss computations in these years. In 2020, the tracer was initialised on 1 April 2020, and was too early. So, we have corrected the simulations with respect to the measurements, and presented the results here. Except these years, tracer simulations for other years are correct and validated. This has been stated in the MS in **lines 82-87, 340-341.**

The black lines drawn only for comparison reasons among the winters. The horizontal and vertical lines are also drawn for the same reason and are the periods of maximum ozone loss and peak chlorine activation days and altitude range. This is mentioned in the figure captions, **lines 323, 341-342.**

Data availability statement: please add doi information or URL for all publicly available data

Done. We have added the url in the Data availability statement, **lines 253-255.**

I don't see where/how you addressed Referee #2's question about mixing up MERRA-2 and ECMWF analysis data. In Lines 64/65 it is stated that MERRA-2 meteorology is used while lines 72/73 state that the REPROBUS model is driven by ECMWF operational analysis. Referee #2 has asked why you did use the meteorology that drives the model also for your analysis and I do not see that question answered, neither in your response nor in the revised manuscript.

Done. Please note that REPROBUS is standard, stabilised, widely-used and well-established model and driven by ECMWF data. The model has been running with the data for years, and is stabilised. We used the MERRA-2 data for meteorology because all required parameters (fluxes, waves, temperature and winds) are available in high resolution, which was not the case for ECMWF (operational data that used in the model). Therefore, to make a comprehensive analysis on the meteorology of the winter, including the status of PSCs, we had to use MERRA-2. Please note that many previous studies also use the MERRA-2 data for meteorological analysis in the polar winters, which would also facilitate the comparison of our results with the previous analysis. Therefore, we used the MERRA-2 data. Hope, the editor and referees agree to that. This is mentioned in **lines 67-70**.

In fact, we have taken the PSC area calculated from MERRA as available from https://ozonewatch.gsfc.nasa.gov/. We apologize for the confusion. This is mentioned in **line 75.**

I don't see a sufficient discussion of the mixing up latitude with equivalent latitude raised by Referee #1

Methods. Since the loss calculation is based on the equivalent latitude (**Line 95 in page 4**), the authors still use the geographic averaged latitude to do other calculations (for example, temperature, PSC etc..). I would suggest the authors use the same criteria to re-make the figures.

Done. Please note that these types of analyses are mostly performed for polar cap temperature, winds, etc, which is why we have presented the analysis this way. These are also needed to check major and minor warming criteria, as we presented in **lines 104–107**. In addition, this is also needed to compare with previous studies (e.g. **lines 106–107**). Therefore, we have kept the original analyses for Figures 1 and 2. However, we respect the referee's comment and we have done the temperature and wind analysis inside the vortex and it is presented in Figure S1 and related text in **lines 140–146.**

choice of words regarding "temperature growth" etc.: please provide an accurate track of changes made for assessment.
Done. We have removed those words. Please see the track-change version of MS. Thank you.